# Protein-Reduced Complementary Foods Based on Nordic Ingredients Combined with Systematic Introduction of Taste Portions Increase Intake of Fruits and Vegetables in 9 Month Old Infants: A Randomised Controlled Trial

**DOI:** 10.3390/nu11061255

**Published:** 2019-06-02

**Authors:** Ulrica Johansson, Inger Öhlund, Olle Hernell, Bo Lönnerdal, Lene Lindberg, Torbjörn Lind

**Affiliations:** 1Department of Clinical Sciences, Paediatrics, Umeå University, SE 901 85 Umeå, Sweden; inger.ohlund@umu.se (I.O.); olle.hernell@umu.se (O.H.); torbjorn.lind@umu.se (T.L.); 2Department of Nutrition, University of California, Davis, CA 95616, USA; bllonnerdal@ucdavis.edu; 3Department of Public Health Sciences, Karolinska Institute and Centre for Epidemiology and Community Medicine, Stockholm County Council, SE 104 31 Stockholm, Sweden; lene.lindberg@ki.se

**Keywords:** infant feeding, Nordic diet, eating behaviour, repeated exposure

## Abstract

Fruits and vegetables are healthy foods but under-consumed among infants and children. Approaches to increase their intake are urgently needed. This study investigated the effects of a systematic introduction of taste portions and a novel protein-reduced complementary diet based on Nordic foods on fruit and vegetable intake, growth and iron status to 9 months of age. Healthy, term infants (*n* = 250) were recruited and randomly allocated to either a Nordic diet group (NG) or a conventional diet group (CG). Infants were solely breast- or formula-fed at study start. From 4 to 6 months of age, the NG followed a systematic taste portions schedule consisting of home-made purées of Nordic produce for 24 days. Subsequently, the NG was supplied with baby food products and recipes of homemade baby foods based on Nordic ingredients but with reduced protein content compared to the CG. The CG was advised to follow current Swedish recommendations on complementary foods. A total of 232 participants (93%) completed the study. The NG had significantly higher intake of fruits and vegetables than the CG at 9 months of age; 225 ± 109 g/day vs. 156 ± 77 g/day (*p* < 0.001), respectively. Energy intake was similar, but protein intake was significantly lower in the NG (−26%, *p* < 0.001) compared to the CG. This lower protein intake was compensated for by higher intake of carbohydrate from fruits and vegetables. No significant group differences in growth or iron status were observed. The intervention resulted in significantly higher consumption of fruits and vegetables in infants introduced to complementary foods based on Nordic ingredients.

## 1. Introduction

Early life is a period of great importance for future dietary choices, since many of the preferences established during childhood remain throughout life [1,2]. Thus, establishing healthy food choices early in life will lead to long-term health benefits [3,4,5]. Consumption of fruits and vegetables is beneficial for a number of long-term health outcomes [6]. However, these foods are under-consumed and intakes do not meet recommendations in most age groups in most settings [7,8,9,10], which contributes significantly to non-communicable diseases such as obesity, type 2 diabetes, cardiovascular disease and cancer [6,11,12,13]. Therefore, increasing the intake of fruit and vegetables is a public health priority [14].

The aetiology of low fruit and vegetable intake among children is multifaceted [15,16,17]. Particularly, introduction of new vegetables may be challenging during certain periods of childhood as children find them difficult to accept [18]. Nevertheless, the so-called high sensitivity period, which lasts from around 3–4 to 12 months of age may be a period when infants are more prone to accept new foods, including bitter and possibly sour flavours [19,20]. 

Several observational studies have shown that repeated exposures of fruits and vegetables during infancy may initiate long-term acceptance of these food items [21,22,23,24]. Many of these studies have been carried out in laboratory settings and have not included homemade baby food [21]. Consuming homemade baby food has been associated with greater dietary diversity and reduced adiposity during the first year of life [25]. We are unaware of any randomised trials in the paediatric population where repeated exposures of fruits and vegetables have been tried within the home setting involving cooking of baby food with the aim to increase fruit and vegetable acceptance during infancy [21]. 

High protein intake in infancy has been associated with high body mass index (BMI) and overweight later in life [26,27]. The protein intake among infants and children in the Nordic countries is high, leading to an increased risk of overweight and obesity [28,29,30,31]. Current dietary recommendations for infants and children suggest lower protein intakes [32] compared to previous guidelines [33]. To achieve this, animal foods, high in protein, may be replaced by vegetables with lower protein content [34], instead increasing intake of carbohydrates from vegetables and fruits.

Studies on the Nordic Diet [35], with its emphasis on intakes of regionally produced fruits, berries, vegetables, tubers, and legumes, higher intakes of whole-wheat, vegetable fats and oils, fish and egg, but less calories from meat have shown several positive health effects among Danish school children [36], and adults from Nordic countries [37,38,39,40,41]. Such a diet may be of particular health benefit if commenced early in life [38], especially if combined with a reduction in total protein intake [31]. However, this is the first study evaluating the Nordic Diet in infants fed complementary foods. 

The specific aims of this study were to (a) investigate the feasibility of systematic introduction of taste portions with repeated exposures of a variety of fruits and vegetables to healthy infants from 4 to 6 months of age, and (b) compare a protein-reduced complementary diet with Nordic foods to the regular Swedish complementary diet on growth, measures of iron status and intake of fruits and vegetables up to 9 months of age.

## 2. Materials and Methods

### 2.1. Study Participants, Design, Allocation and Blinding

All parents to healthy, term, 4 month old singletons from Umeå were invited by letter to participate in the study between April 2015 and January 2018 (ClinicalTrials.gov registration number NCT02634749). Umeå is a university town with 125,000 inhabitants in the north of Sweden. Of the 540 families that showed interest in participating, 250 were recruited (Figure 1). In the present study, we used a portfolio design, meaning that we applied a number of interventions to the same group. A detailed description of the study protocol has been published elsewhere [42]. Inclusion criteria were healthy, singleton infants, 4–6 months of age, born after >37 weeks of gestation and birth weight >2500 g living in Umeå and that the participants would remain in the study area (Umeå municipality) and would not commence childcare outside the home during the extent of the study. Exclusion criteria were chronic illnesses that would affect nutrient intake or the outcomes of the study, iron deficiency or any other biochemical abnormality, or having started feeding complementary foods at the time of recruitment. Infants were solely breast-fed and/or formula-fed at study start. When parents considered it appropriate to introduce complementary foods, their infant was randomly allocated to one of two study groups, i.e., the Nordic group (NG, intervention) or the conventional group (CG, control). Parents, the research nurses and the dieticians could not be completely blinded to the group allocation since the food items given to the infants could not be blocked out entirely. Other staff, including the laboratory team and the researchers responsible for the analyses were blinded to the participants’ group allocation.

### 2.2. New Nordic Food and Diet

The food choices in the NG were based on the New Nordic Food Manifesto from the Nordic Council of Ministers [43,44], which promotes Nordic-grown, season-based regional foods that are rich in fish, plant foods and vegetable oils and recommends reduction of added sugar, saturated fat, meat and meat products. The nutrient composition of the New Nordic Diet complies with the Nordic nutrition recommendations [45]. In this study, the Nordic diet (ND) also included a reduced overall intake of protein and systematic introduction of fruit and vegetables with repeated exposures during the very first weeks of complementary feeding (CF).

### 2.3. Introduction of Taste Portions in the Nordic Group

When the participants in the NG (*n* = 125) started CF, they commenced 24 days of systematic taste portions (Figure 2). Parents decided themselves when to start these taste portions when the infants were between 4 and 6 months of age, but not later than at 6 months. As these taste portions were not commercially available, the parents were provided with recipes for homemade fruit and vegetable purées produced specifically for the study, in total eight different recipes. Parents purchased all the ingredients for the recipes and prepared the purées themselves. Parents could choose which liquid to use in the preparation, i.e. either breast milk, formula or oat cream. The fruit and berry purées contained 80%–89% (w/w) raw material, 2%–7% (w/w) breast milk, formula or oat cream, 0%–12% (w/w) added sugar and 0%–13% (w/w) added water. Sugar was added to the purées in different amounts to make them more palatable, from 3% (lingonberry and raspberry, respectively) to 9% (buckthorn) and 12% (cranberry). The vegetable purées contained 70%–81% (w/w) raw material, 8%–16% (w/w) breast milk, formula or oat cream, 0%–14 % (w/w) added water, 1.3%–1.8% (w/w) added herbs (parsley or dill), 1.5%–1.8% (w/w) added rapeseed oil and 0.6%–0.8% (w/w) added butter. 

### 2.4. Adherence to the Taste Portions in the Nordic Group

The systematic taste portion schedule consisted of three days per fruit (four in total) or vegetable (four in total); the taste portions were offered as 5–15 mL of purée up to three times per day, in total nine exposures for each purée. The taste portion schedule was designed with the intention to first introduce sweet tastes, e.g., pea purée and then move on to more bitter and sour tastes resulting in 72 exposures over 24 days (Figure 2). Sweet tastes are inherently familiar to infants, which made them a natural first step before moving on to more bitter and sour tastes, which require some introduction [46]. Parents were instructed how to feed the infant with the daily taste portion besides breast-feeding or formula-feeding and how to report acceptance or refusal behaviour. Each day of the taste portion schedule parents answered an electronic questionnaire (Textalk Web survey, Sweden) [47]. Questions included which fruit or vegetable used on that particular day, the number of exposures that day, which liquid (breast milk, formula or oat cream) was used to prepare the purée, amount of purée consumed (in teaspoons) and number of refusals the child made when offered the purée. Infants’ refusal behaviour was checked off with yes or no per each exposure. An accepting behaviour was counted if the child had the purée in the oral cavity irrespective whether the child swallowed the portion or not.

### 2.5. Support through Social Media in the Nordic Group

Parents in the NG group were invited to participate in a closed Facebook group. The purpose of this group was to inspire and support the parents through images, videos and messages to prepare taste portions of purées and other recipes provided in the study, and to enhance the parents’ willingness to complete the taste portion schedule. The videos illustrated how to prepare the purées in the taste portion schedule. The research dietician (UJ) recorded all the videos. The group also enabled chats and possibilities to put forward questions from the participants. The research nurses and the doctoral student (UJ) were responsible for the content in the Facebook group and responded to all questions.

### 2.6. Baby Food Recipes in the Nordic Group

The NG received a folder with baby food recipes to encourage preparation of homemade main course meals with a higher content of vegetables in comparison to commercial baby food jars. The folder included 28 recipes of homemade baby food prepared specifically for the study with four 7-day nutritionally balanced menus (lunch/dinner) based on the ND concept, but also reduced in protein (1.8–2.9 g protein per 100 g) compared to commercial baby food in glass jars (BIG) and pouches (2.4–4.0 g). The reduced amount of protein in the homemade baby food was replaced with a higher content of vegetables and therefore higher content of carbohydrate, whereas the content of fat and total energy content remained the same. The group was also supplied with 28 additional recipes, designed specifically for the study, for fruit, berry and vegetable purées apart from those eight that were included in the taste portion schedule. To help parents navigate among the different purées, they received a food map with images of the in total 28 vegetables, berries and fruits used in the study, to enhance the motivation to feed the infant using the study recipes of Nordic food.

### 2.7. Baby Food Products in the Nordic Group

From 6 months of age, after the taste portion schedule was completed, the NG was supplied with age-appropriate BIG and pouches with Nordic ingredients. Part of the protein source in the BIG was replaced by a higher content of vegetables from either vegetable purées in glass jars or homemade vegetable purées made by the parents from the received folder with information about adding extra fat from rapeseed oil to provide sufficient energy density to the food. The group also received age-appropriate, iron-fortified porridge and milk cereal drink reduced in protein content per 100 g by approximately 30%. Participants were encouraged to use the study products with Nordic ingredients together with the recipes for homemade baby food.

### 2.8. The Conventional Group and Taste Portion Introduction

Participants in the CG (*n* = 125) received a brochure from the Swedish National Food Agency [48] which contained written, parent-oriented information on the Swedish recommendations for taste portions and solid food introduction during infancy. Parents decided themselves when to start with the first taste portions between 4 and 6 months of age. Similar to the NG, participants were either exclusively breast-fed or formula-fed at the study start and had not been fed taste portions or other solid foods. The national nutritional recommendations suggest 6 months of exclusive breast-feeding or formula if the child is not breast-fed. If the child is very interested in other foods, the proposal is to offer tiny taste portions besides breast-feeding or formula-feeding, starting no earlier than 4 months of age. These tiny taste portions are intended to be family foods in small amounts offered from the parent’s fingers to the baby mouth. The recommended amount to begin with is about 1 mL, which should slowly increase to a couple of teaspoons (10 mL) during the following couple of weeks beside breast-feeding or formula-feeding. From 6 months of age, the guidelines recommend [48] offering taste portions of potatoes, rice, vegetables (broccoli, cauliflower, green peas, corn), root vegetables (carrot, parsnip) and fruits (banana, apple, pear). In the CG, no further instructions were given from the research team besides the official National recommendations regarding complementary feeding and weaning. However, the participants had, as any family, full access to online information from the Swedish Food Agency website as well as support free of charge from the community child health care centres. 

### 2.9. Baby Food Products in the Conventional Group

The CG was supplied with age-appropriate BIG and pouches from 6 months of age. These products were the same as those commercially available on the Swedish market (Semper AB) with no modification of the protein content. The participants also received age-appropriate, iron-fortified porridge and milk-cereal drinks of the same type that is commercially available on the Swedish market with no modification of protein content. Besides these, the participating families were free to use whichever baby food they found to their liking. Depending upon type of BIG the amounts of vegetables varied from 4% to 60% (w/w).

### 2.10. Food Records and Dietary Assessment 

Five-day food records (FR) were collected at 6 and 9 months of age. Parents were asked to record everything their child ate and drank, including breast milk and any food supplements, e.g., vitamins, using a pre-printed FR, within two weeks of the infant’s 6- and 9-month birthday. Each day of the FR, parents noted meal type, time of day, which foods and drinks their infant was offered including amounts and brand names. Breast milk intake was recorded, at the discretion of the mother, as ‘meals’ or ‘snacks’ estimated to 102 or 25 g of milk, respectively [49,50]. The reported food and drink intake was converted to grams using standardised weights for consumed foods from the Swedish Food Agency Database [51]. To calculate mean daily energy intake (EI), macronutrient sub-classes and fruit and vegetable content, the software Dietist Net Pro (Kost och Näringsdata AB, Sweden) and the food composition database (version 01/01/2017) from the National Food Administration, Sweden [51] were used. The database was complemented with special products for infants used in the study with nutrient contents analysed and supplied by Semper AB.

Two trained paediatric dieticians calculated mean daily energy intake (EI), macronutrient sub-classes, and fruit and vegetable content from the FR. Fruit juices, vegetables juices, potatoes, chili, garlic, ginger and herbs were not included in the dietary assessment for fruits and vegetables intake. Data on fruit and vegetable content in the study products was supplied by Semper AB. Content in other baby food brands and products were calculated from data provided from these companies’ websites.

### 2.11. Anthropometry

Infants were measured at the paediatric research facility at Umeå University Hospital at baseline and 9 months of age. Anthropometric data were collected according to standardised procedures [52]: nude weight was measured to the nearest 5 g using electronic scales (Seca 727, Seca, Hamburg, Germany), recumbent length was measured to the nearest 0.1 cm using an infantometer (Seca 416, Seca, Hamburg, Germany) and head circumference measured to the nearest 0.1 cm using a non-stretchable measuring tape (Seca 212, Seca, Hamburg, Germany). Body mass index (BMI) was calculated as weight in kg divided by (length in m)^2^. Weight-for-age, length-for-age, BMI-for-age and head circumference-for-age *z*-scores were calculated according to the World Health Organization Child Growth Standards [53].

### 2.12. Blood Samples and Laboratory Analyses

Venous blood samples were collected by experienced paediatric nurses after a 2-h fast at baseline and at 9 months of age in an EDTA-containing tube and a serum separator tube. If the child was ill or had recently been immunised, sampling was postponed by 2 weeks to avoid the influence of an acute-phase response on blood indices. Haemoglobin mean corpuscular volume, s-ferritin, s-iron, s-transferrin, s-folate and s-urea were analysed within 4 h of collection at the Department of Clinical Chemistry, Umeå University, Sweden. Haemoglobin and mean corpuscular volume were measured in whole blood using a Sysmex XN-9000 Automated Haematology Analyser (Sysmex Corporation, Japan). Blood in the serum separator tube was centrifuged for 10 min at 2000 rpm, and the separated serum was analysed for ferritin, iron, transferrin, folate and urea using a Roche Cobas 8000 (Roche Diagnostics). S-iron and transferrin concentrations were used to calculate transferrin saturation using the following formula: transferrin saturation (%) = s-iron (μmol/L)/(s-transferrin (g/L) × 25.1) × 100 [54].

### 2.13. Demographic Variables

Demographic information was collected from both caregivers through an electronic survey (Textalk Web survey, Sweden) [47]. The questionnaire was answered when the child was 4–6 months of age. The survey included information on current employment, education level, health factors such as smoking and food allergies, age of parents, family composition, income, ethnicity and number of siblings.

### 2.14. Group Size Calculation

Sample size calculation for the study was based on the detection of a difference in body fat mass between the Nordic and Conventional groups of 0.4 SD at 12 months of age, with a power of 80% and α set to 0.05. Allowing for an attrition rate of 20%, we recruited 125 participants per study group.

### 2.15. Ethical Considerations

The study was approved by the Regional Ethical Review Board at Umeå University (2014-363-31M), Umeå, Sweden. Written informed consent was obtained from both caregivers.

### 2.16. Statistical Analyses

Statistical analyses were performed using SPSS 24.0 (SPSS, Chicago, IL, USA). Results are presented as means (± standard deviations, SD) or if non-parametric data as median (min–max and IQR, interquartile range). Categorical data are presented as numbers and percentages. Energy and macronutrient intake are expressed as kilojoules (kJ) and grams (g) per day, respectively. EI per kg body weight was calculated as kilojoules (kJ) and protein (g) per kg body weight per day. Fruit and vegetable intakes are expressed as g per day. The significance level was set at *p* < 0.05. For comparison between the groups, independent *t*-test was used on normally distributed data and the Mann–Whitney test was used for non-normally distributed data. Chi^2^ test was used for comparisons between categorical variables. For comparisons within the NG for the taste portion schedule for type of foods, amounts per food item, refuses per food item and total exposures, ANOVA was used. In the statistical analysis of the main course meals groups, the independent Kruskal Wallis Samples *t*-test for non-parametric data was used. Effect size for parametric data was calculated with Glass’s delta [55] (Δ), where $\Delta =\frac{\left(Mean_{1}-Mean_{2}\right)}{s.d.control}$. Glass´s delta calculation was used when it was more representative SD of a population from the CG comparing to the children in the NG affected by an intervention. The Glass’s delta interpretation of a large effect size (*r =* 1) means that 84th percentile of the CG distribution is the mean value of the NG.

## 3. Results

### 3.1. Study Participants

Of the 250 recruited infants, 232 (93%) completed the study until 9 months of age. There were no differences at baseline in anthropometric and demographic variables between the study groups (Table 1). Attrition was significantly higher in the NG, 14 vs. four dropouts, in NG and CG, respectively (Chi^2^
*p* = 0.014). No differences were found between dropouts and those who remained in the study for any anthropometric, health or demographic variables. 

### 3.2. The Taste Portion Schedule in the Nordic Group

Of the 125 infants in the NG, 95% completed the taste portion schedule between 4.8 and 5.7 months of age. During the 26.7 ± 7.5 days of the schedule, the participants experienced 63.6 ± 9.3 exposures of either fruit or vegetable purée, which was 89% of total possible exposures. Parents recorded 3.2 ± 4.3 (5%) refusals during the same time. Mean daily intake of the purées was 22.7 ± 12.1 g. No significant differences were found in food acceptance between type of fruit or vegetables, amounts of eaten food or number of refusals (Table 2). The optional fluid added to the taste portions recipes varied and were categorised into breast milk 42% (*n* = 50), formula 38% (*n* = 45), mix of breast milk and formula 19% (*n* = 23), and oat cream 1% (*n* = 1). During the taste portion schedule 75% (*n* = 89) of the infants were breast-fed. Among the participating families, 75% (*n* = 89) joined the Facebook group, whereof 29% (*n* = 26) had both caregivers registered in the group. 

### 3.3. Energy and Macronutrient Intake and Breast-Feeding Duration 

No significant differences in mean daily energy or fat intake were found between the study groups at either 6 or 9 months of age (Table 3). There was, however, a significantly higher daily intake of total protein at both 6 and 9 months of age and a higher protein intake per kg body weight at 9 months in the CG group (Table 3). The differences expressed in effect size (Glass’s ∆ 0.91) were large. In both groups, however, protein intake was within recommended levels. At 9 months, but not at 6 months, mean daily intake of carbohydrates was significantly higher in the NG compared to the CG. Breast-feeding prevalence did not differ between the groups at 6 and 9 months of age. In fact, at 6 months, 73% and 69% of infants were breast-fed in the NG and CG, respectively and at 9 months 38% were breast-fed in both groups.

### 3.4. Fruit and Vegetable Intake

At 6 months of age, there were no differences between the groups in fruit and vegetable intakes, either separately or in combination, but at 9 months of age, there were clear differences (Figure 3a): intake of fruit and vegetables were 44% higher in the NG, both separately and combined. The difference in fruit and vegetable intake (225 ± 109 g/day vs. 156 ± 77 g/day in the NG and CG, respectively) (Figure 3b) showed a large effect size (Glass’s ∆ 0.91). However, the origin and varieties of fruits differ between study groups, where intake of exotic fruits, which included banana, mango, orange, mandarin, peach and melon were higher in the CG (Figure 3c), whereas intake of berries and root vegetables was significantly higher in the NG (Figure 3c,d). 

### 3.5. Main Course Meals

In both study groups, two out of three main course meals, i.e., lunch/midday meal and supper/evening meal were served as BIG. However, families in the NG served homemade baby food significantly more often compared to the CG (0.3 ± 0.5 meals/day vs. 0.0 ± 0.2 meals/day, respectively, *p* < 0.001). Infants in the CG were instead served family foods significantly more often as main course meals compared to the NG (0.2 ± 0.4 meals/day vs. 0.0 ± 0.2 meals/day, *p* < 0.001). Porridge, cereal drink, breast milk and/or formula were served to lunch/midday meal in the same frequency in both groups (0.2 ± 0.4 meals/day vs. 0.3 ± 0.4 meals/day, respectively).

### 3.6. Anthropometry and Biochemistry

There was no difference in weight, length and head circumstance between the groups at 9 months of age (Table 4), nor were significant differences found in growth rate between baseline and the 9-month measurements. Laboratory markers showed no significant differences in haemoglobin, mean corpuscular volume or iron status between the two groups (Table 4). S-urea was significantly lower (*p* < 0.001) and s-folate was significantly higher (*p* = 0.05) in the NG compared to CG while no other differences in growth, anthropometrics or biomarkers were found between study groups (Table 4). 

## 4. Discussion

This is the first study to evaluate a protein-reduced complementary diet based on Nordic foods in a randomised trial. We show that this intervention, where we also took advantage of the high sensitive period in early infancy to systematically introduce a wide variety of bitter and sour flavours, resulted in a higher intake of fruits and vegetables at 9 months of age without negative effects on infant growth or biomarkers of iron status. We also show that using ND in complementary feeding is feasible and associated to environmentally sustainable food choices [35,56]. Early introduction, i.e., already during infancy, of these foods may contribute to enduring behavioural changes [57,58,59,60] towards healthier eating habits among children in Sweden and elsewhere. In order to evaluate possible long-term effects of ND in the present study, we will continue to follow the participants into childhood. 

Participants allocated to the NG adhered very well to the taste portion schedule, despite a large variation in flavour, including some very bitter and sour experiences, which infants are normally not exposed to. This systematic introduction of taste portions in combination with several other efforts, i.e., provision of recipes to prepare homemade baby food, supplying participants with specific baby food products and support to the parents/caregivers through social media resulted in significantly higher intake of fruits and vegetables, especially from the Nordic region, at 9 months compared to the group provided current national recommendations regarding complementary feeding and regular baby food products. The NG also had a reduced protein intake, i.e., by 26% compared to the CG at 9 months. The lower protein intake was biochemically confirmed by a significantly lower s-urea concentration compared to the CG. A reduction in protein intake of 26% in this age group is substantial and in line with our objectives for this study. Although longer term studies are needed to evaluate potential effects on risk of diabetes and obesity, a reduction of serum urea is usually correlated with a reduction in the insulinogenic branched-chain amino acids. To reach a similar energy intake as the CG, the NG increased their carbohydrate intake, particularly from fruits and vegetables, which in turn was reflected by higher s-folate in the NG. Growth was similar in the two groups and we did not detect any significant differences in iron status or risk of developing iron-deficient erythropoiesis at the 9 months follow-up.

This study confirms and extends the experience from previous studies on early introduction and repeated exposures of fruits and vegetables in infancy [22,24]. In addition to earlier efforts, we could demonstrate these effects in a home-based setting, over an extended time period and within a randomised trial [21]. Moreover, in the present study we were able to also assess the infants’ diet including total daily intake of fruits and vegetables, which is in contrast to previous studies with repeated exposures and follow-up studies [20,22,24,61]. In a study from Australia, Campbell et al. reported a randomised, parent-focused, intervention study on infants’ obesity risk behaviours and BMI. The authors found no effect on daily intakes of fruits and vegetables at 9 months of age, when the intakes were assessed from three days of 24-h diet recalls [62]. Similar to the present study, intakes of fruits and vegetables were not normally distributed with wide variations in daily intake. Compared to the present study, vegetable intake at 9 months of age was higher in both the intervention and the control group in the Australian study. Mean fruit intake was higher in the NG in the present study compared to the intervention group in the study by Campbell et al. [62]. The differences in outcomes may be partly explained by differences in measurements and calculations of intake of fruits and vegetables, but also by the difference in the chosen intervention strategy and maybe food culture differences.

### Strengths and Weaknesses

Due to the portfolio design of our study, we cannot point out which part of the intervention was the most efficient in reaching the outcome. Both groups showed excellent compliance to the protocol. In the NG, this included procuring and preparing the foodstuffs needed to make homemade baby food. Homemade baby food has been associated with higher dietary diversity and lower adiposity during infancy [25]. We encouraged the use of such homemade baby foods in conjunction with age-appropriate, iron-fortified, industrially manufactured baby porridge and milk cereal drink. Our study thus provided conditions to increase the knowledge about the domestic food culture, a chance to learn about foods grown and produced in the country and how to use this produce in the preparation of homemade baby foods and ultimately for the whole family. Anecdotally, some families reported that the whole family started to eat Nordic food during the study. Other strengths include that the taste schedule demanded daily reporting, which reduces the bias from self-reported data. The intervention was not without challenges and the NG had significantly higher attrition compared to the CG, with seven participants declining continued involvement due to the ND. The higher attrition from the NG was, however, not dependent on socioeconomic factors such as educational level, ethnicity or if the caretaker was a single parent and we could see no other systematic differences between those who left and those who remained in the study. The extra support given to the NG in the form of social media encouragement and specific recipes of homemade baby foods did not prevent the difference in attrition between the groups, and the divergence would maybe have been even greater without the assistance given. This may be important to consider in future diet intervention trials. In other trials, the attrition rate has been lower [62,63] and in comparison to this study it may be explained by the strict portfolio diet with more workload for the families in the NG. 

Other limitations of the present study include lack of comprehensive data on exactly which type of taste portions were given in the CG, only that they were advised to start complementary feeding according to current Swedish recommendations. A previous study [64] evaluated introduction age and volumes of taste portions among Swedish infants and confirmed them to be according to recommendations. Most of the infants were introduced to taste portions at 4–6 months of age, only 5% after >6 months of age and volumes for a “taste portion” of at least 10 mL were reported from 76% of the participants [64]. In accordance with our findings from the taste schedule, Hörnell et al. found that introduction of taste portions was a lengthy process among breast-fed infants, and median duration was 28 days from first taste portion to meal consumption (>10 mL) [50]. The participants in the NG completed the taste portions schedule within 26.7 ± 7.5 days, more days than expected from the 24-days taste schedule. The taste portions consumption in the NG were higher (daily mean intake 22.7 ± 12.1 g) in comparison to Hörnell et al and can be explained by the mixed group of both breast-fed (75%) and formula-fed (25%) infants and the intervention requested them to offer at least 15 g of taste portions per day. 

Participating parents were highly educated, which may affect external validity. Finally, there may have been unknown variations in flavour and consistency of the taste portions among the randomised participants in the NG due to differences in their preparation but also due to seasonal changes in the various fruits and vegetables. Other studies have used foods prepared in the laboratory or by a baby food manufacturer, which may ensure their conformity [24,61]. However, we believe that the strategy employed in the present study better reflects a real-life situation and introduces important practices for the parents in preparing homemade baby foods.

In conclusion, we show that infants provided with a protein-reduced complementary diet based on Nordic foods including a systematic introduction of taste portions from 4–6 months of age improved their intake of fruits and vegetables and established eating behaviours that remained until 9 months of age. The study contributes new information on feasible, home-based interventions to advance our knowledge surrounding complementary feeding towards healthier and more environmentally sustainable food choices and will thus improve future nutritional guidelines for infants in Sweden and elsewhere. However, assessment of long-term effects of ND need continued follow-up. 

## Figures and Tables

**Figure 1 nutrients-11-01255-f001:**
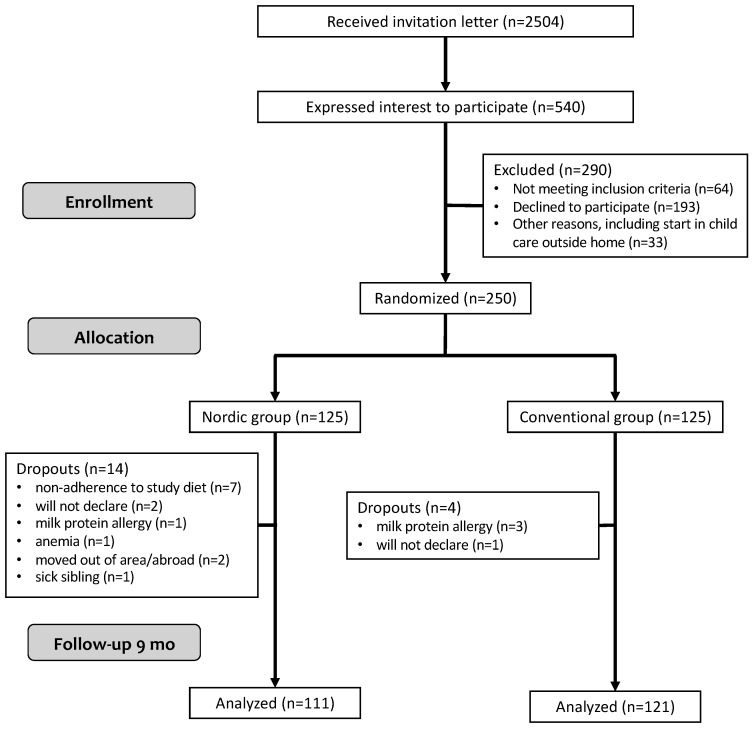
Flowchart diagram of the Optimised complementary feeding study.

**Figure 2 nutrients-11-01255-f002:**
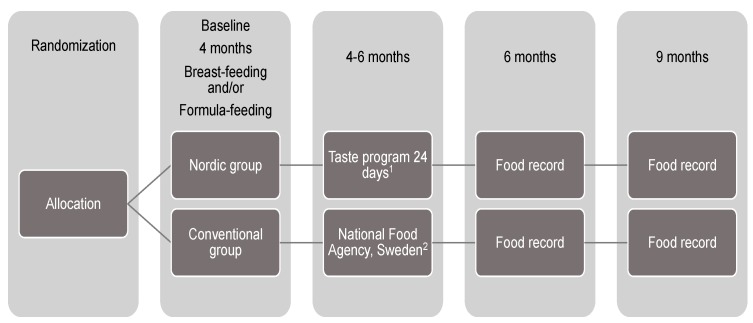
Flow diagram of procedure in the Optimised complementary feeding study.^1^ Parents in the Nordic group were provided with home-made Nordic baby food recipes to make 4 vegetable purees and 4 fruit/berry purees which were fed to the infants three times per day on designated days of the 24-day exposure period, in total 72 exposures. ^2^ Parents in the Conventional group received a brochure from Sweden Food Agency; How to feed their infant during the first year. The group received no further instructions from study personnel on when or how to feed their infants.

**Figure 3 nutrients-11-01255-f003:**
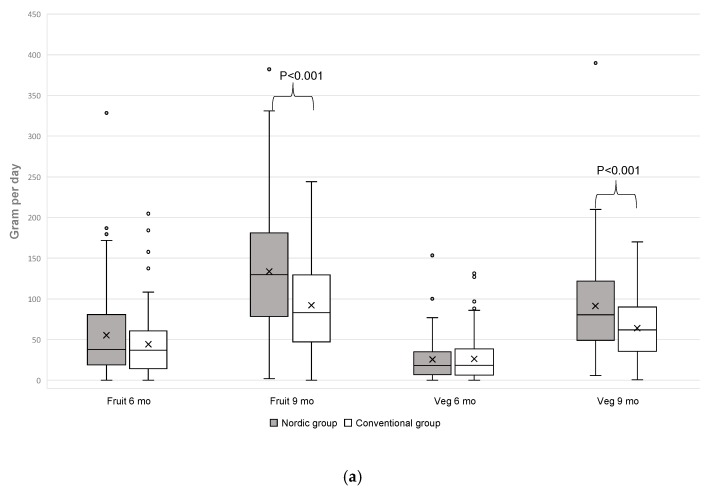
(**a**) Daily intake of total fruits and vegetables (g) at 6 and 9 months of age, (**b**) Daily mean intake of total fruits and vegetables (g) at 9 months of age, (**c**) Daily intake of berries, exotic fruits and other fruits at 9 months of age, (**d**) Daily intake of roots and other vegetables at 9 months of age. Fruits juices, vegetables juices, potatoes, chili, garlic, ginger and herbs are not included in the assessment.

**Table 1 nutrients-11-01255-t001:** Baseline data of OTIS study (*n* = 250).

	Nordic Group *n* = 125	Conventional Group *n* = 125
Girls/boys (*n*, %)	52/73 (42)/(58)	56/69 (45)/(55)
	**mean ± SD**	**mean ± SD**
Age at baseline (months)	4.5 ± 0.5	4.5 ± 0.5
***Neonatal characteristics***		
Birth weight (kg)	3.6 ± 0.4	3.6 ± 0.5
Birth length (cm)	50.7 ± 1.9	50.6 ± 2.1
Birth head circumference (cm)	35.4 ± 1.3	35.5 ± 1.4
Gestational age (weeks)	39.6 ± 1.3	39.8 ± 1.3
***Anthropometry***		
Weight (kg)	7.3 ± 0.8	7.3 ± 0.8
Weight-for-age *z* score	0.31 ± 0.87	0.35 ± 0.89
Weight-for-length *z* score	0.07 ± 0.98	0.12 ± 0.96
Length (cm)	65.2 ± 3.0	65.1 ± 2.4
Length-for-age *z* score	0.47 ± 0.85	0.50 ± 0.94
Head circumstance (cm)	42.4 ± 1.3	42.5 ± 1.3
Head circumstance-for-age *z*-score	0.69 ± 0.89	0.75 ± 0.90
Body mass index (BMI; kg/m^2^)	17.2 ± 1.5	17.2 ± 1.4
BMI-for-age *z* score	0.05 ± 0.96	0.09 ± 0.95
***Breast-feeding***	***n* (%)**	***n* (%)**
Breast-feeding	95 (76)	94 (75)
Never breast-fed	2 (1.6)	2 (1.6)
	**mean ± SD**	**mean ± SD**
Duration of exclusive breast-feeding (months)	4.1 ± 1.5	4.2 ± 1.4
***Laboratory markers***	**mean ± SD**	**mean ± SD**
Haemoglobin (g/L)	116 ± 7	116 ± 8
Mean corpuscular volume (fL)	77.7 ± 3.3	77.7 ± 3.0
S-Transferrin (g/L)	2.3 ± 0.3	2.3 ± 0.4
Transferrin saturation (%)	16.5 ± 5.5	16.3 ± 6.2
S-Ferritin (μg/L) (geometric mean)	107.6 ± 2.1	105.7 ± 2.1
S-Urea (mmol/L)	2.0 ± 0.5	2.0 ± 0.5
S-Folate (nmol/L) (median, IQR)	44.0 (38–45)	41.0 (36.8–45)
***Family characteristics***	**mean ± SD**	**mean ± SD**
No siblings (*n*, %)	63 (52)	69 (55)
Mothers’ age (years)	31 ± 4.6	31 ± 4.9
Partners’ age (years)	34 ± 5.0	32 ± 5.4
***Education level Mother***	***n* (%)**	***n* (%)**
Elementary school	2 (1.6)	4 (3.2)
High school	34 (27.9)	36 (28.8)
University	86 (70.5)	84 (67.2)
***Education level Partner***		
Elementary school	10 (8.2)	9 (7.2)
High school	44 (36.1)	48 (39.2)
University	68 (55.7)	67 (53.6)
***Ethnicity (born in Sweden)***		
Mother	120 (98.4)	118 (95.2)
Partner	110 (88.0)	104 (83.2)
***Health***		
Smoking mother	2 (1.6)	1 (0.8)
Smoking partner	13 (10.7)	13 (10.4)
Food allergy mother (self-estimated)	25 (20.3)	20 (16)
Food allergy partner (self-estimated)	13 (10.6)	17 (13.6)
Food allergy siblings	4 (7) ^1^	5 (9) ^1^
***Annual income per household (Euro €/per thousand)***		
<10:	1 (0.8)	2 (1.6)
10.0–19.9:	8 (6.6)	10 (8.0)
20.0–29.9:	30 (24.6)	32 (25.6)
30.0–39.9:	35 (28.7)	34 (27.2)
40.0–49.9:	30 (24.6)	30 (24.0)
50.0–59.9:	11 (9.0)	8 (6.4)
>60:	7 (5.7)	7 (5.6)

^1^ Of 59 participants with siblings in the NG and 57 participants with siblings in the CG.

**Table 2 nutrients-11-01255-t002:** Descriptive data from the taste portion schedule in the Nordic group (*n* = 119).

Taste	Total no. of Exposures (*n*)	Total Amounts of Purée (g)	Amounts of Purée per Exposure (g)	Total no. of Refuses (*n*)
	mean ± SD	mean ± SD	mean ± SD	mean ± SD
Apple	8.3 ± 1.3	72.1 ± 51.2	8.6 ± 5.7	0.3 ± 1.0
Green peas	8.1 ± 1.3	68.7 ± 45.9	8.5 ± 5.3	0.4 ± 0.9
Raspberry	8.0 ± 1.4	67.9 ± 47.2	8.4 ± 7.1	0.5 ± 1.4
Cauliflower	7.9 ± 1.5	76.3 ± 46.7	9.6 ± 5.4	0.4 ± 1.4
Buckthorn/Lingonberry ^1^	8.0 ± 1.7	68.4 ± 47.0	8.3 ± 5.1	0.4 ± 1.0
Turnip	7.9 ± 1.7	77.4 ± 44.6	9.7 ± 5.1	0.3 ± 0.7
Cranberry	7.7 ± 1.7	63.9 ± 44.8	8.2 ± 5.4	0.5 ± 1.4
White radish	7.8 ± 2.0	73.3 ± 46.7	9.4 ± 5.4	0.4 ± 1.2

^1^ Buckthorn berry and lingonberry were used interchangeably.

**Table 3 nutrients-11-01255-t003:** Daily intake of energy and macronutrient at 6 and 9 months of age in the OTIS study.

Intake	Nordic Group Mean ± SD	Conventional Group Mean ± SD	*p* for Difference ^1^
**Energy and macronutrient at 6 months**			
Age at follow-up (months)	6.3 ± 0.6	6.2 ± 0.5	0.30
Energy (kJ)	2941 ± 525	2941 ± 497	0.76
Protein (g)	11.7 ± 3.0	13.1 ± 4.0	0.003
Fat (g)	32.3 ± 6.0	33.1 ± 6.7	0.38
Carbohydrate (g)	87.6 ± 20.9	83.1 ± 16.8	0.07
**Energy and macronutrient at 9 months**			
Age at follow-up (months)	8.7 ± 0.4	8.7 ± 0.3	0.99
Energy (kJ)	3472 ± 498	3432 ± 538	0.56
Energy/bodyweight (kJ/kg)	379 ± 62	372 ± 58	0.38
Protein (g)	15.9 ± 3.6	21.5 ± 5.7	<0.001
Protein/bodyweight (g/kg)	1.7 ± 0.4	2.3 ± 0.6	<0.001
Fat (g)	32.8 ± 5.4	33.7 ± 6.9	0.27
Carbohydrate (g)	113.6 ± 20.6	103.4 ± 18.5	<0.001

^1^ Independent samples *t*-test.

**Table 4 nutrients-11-01255-t004:** Anthropometrical and biochemical outcomes at 9 months of age in the OTIS study.

	Nordic Group Mean ± SD	Conventional Group Mean ± SD	*p* for Difference ^1,2^
***Anthropometry***	***n* = 108**	***n* = 122**	
Age at follow-up (months)	8.7 ± 0.4	8.7 ± 0.4	0.97 ^1^
Body weight (kg)	9.2 ± 1.0	9.3 ± 1.0	0.75 ^1^
Weight-for-age *z* score	0.62 ± 0.91	0.68 ± 0.91	0.60 ^1^
Weight-for-length *z* score	0.60 ± 0.97	0.60 ± 0.96	0.80 ^1^
Body length (cm)	71.8 ± 2.1	72.0 ± 2.5	0.51 ^1^
Length-for-age *z* score	0.41 ± 0.79	0.57 ± 0.99	0.24 ^1^
Head circumstance (cm)	45.7 ± 1.3	45.7 ± 1.5	0.87 ^1^
Head circumstance-for-age *z*-score	0.99 ± 0.87	1.05 ± 1.01	0.82 ^1^
BMI (kg/m^2^)	17.9 ± 1.5	17.8 ± 1.5	0.97 ^1^
BMI-for-age *z* score	0.51 ± 0.98	0.49 ± 0.98	0.92 ^1^
***Laboratory markers***	***n* = 101**	***n* = 114**	
Haemoglobin (g/L)	114.7 ± 6.4	116.3 ± 7.5	0.10 ^1^
Mean corpuscular volume (fL)	75.7 ± 2.5	75.8 ± 2.7	0.69 ^1^
S-Transferrin (g/L)	2.54 ± 0.31	2.58 ± 0.34	0.40 ^1^
S-Transferrin saturation (%)	13.8 ± 5.9	14.4 ± 5.2	0.40 ^1^
S-Ferritin (ug/L)	40.7 ± 1.82 ^3^	40.2 ± 1.90 ^3^	0.88 ^1^
S-Urea (mmol/L)	2.33 ± 0.66	3.26 ± 0.87	<0.001 ^1^
S-Folate (nmol/L)	42 (37.8–45) ^4^	39 (34.8–45) ^4^	<0.05 ^2^

^1^ Independent samples *t*-test, ^2^ Mann–Whitney test, ^3^ geometric mean. ^4^ median (IQR).

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
