# Peer review of "Protein-Reduced Complementary Foods Based on Nordic Ingredients Combined with Systematic Introduction of Taste Portions Increase Intake of Fruits and Vegetables in 9 Month Old Infants: A Randomised Controlled Trial"

_nutrients, 2019, doi:10.3390/nu11061255_

Round 1
Reviewer 1 Report
Scientifically sound carried out study. My main concern with the design is with the lack of support given the control group. In a sense they are almost discarded. Who can tell what would have happened if the control group received the same encouragement as the Nordic group. It is possible parents/caregivers could pick up the shift in attitude. The other concern I have is with the assumption of the findings going into childhood (line 332-334). There are many examples of studies finding something important in early infancy that get overtaken by other factors as the child gets older.
One could argue this is an experiment designed to assess the level of support given to parents. What will happen once the parents/caregivers are not given free food? What will they choose to feed their infants? As well feeding an infant pureed foods ensures they get the food. I accept the changes seen at 9 months however once the infant turns into a child they will make their own choices. Th e concept is important and This requires further study. Hopefully the authors plan to do this.
Author Response
Dear Reviwer,
Thank you for the comments and suggestions to improve the manuscript. In the attached PDF file you find a point-by-point respond to your comments.

Reviewer 2 Report
This study describes a comparison of infants’ intake of fruit and vegetables following adherence to one of two diet types. Though the topic is of importance, much more detailed information should be supplied in the introduction and methods so the reader understands the study objectives and why the specific approach was taken. For example, I remain confused about why the taste portion methodology was so different between the treatments (ie. giving 5-15 ml of puree in teaspoons in the Nordic group vs 1 ml offered from the parent’s fingers in the Swedish group), and why groups were treated differently (ie. NG group received social media support but the CG group did not). I am concerned that some of the differences between the diet groups themselves may have resulted in the reported outcomes. The authors should edit carefully for detail and clarity to explain thoroughly. Minor comments are below.
Line 42: Does under-consumption of vegetables really contribute significantly to obesity, type 2 diabetes, CVD, etc? I would be careful with wording here. For example, isn’t the primary contributor to development of obesity actually an excessive intake of energy and not low intakes of vegetables?
Line 48: I think I know what you’re trying to say in this sentence, but please edit for clarity.
Line 53: may increase instead of may introduce?
Line 54: An explanation of why a focus on homemade instead of store-bought baby foods is important would be welcome.
Line 55: Suggested edit: ‘We are unaware of any’ instead of ‘there are no for us known’
Line 58: High protein intake is often used to manage weight. Can you expand on the link between high protein intake in infancy and overweight so this is clear?
Line 61: Can you further explain the New Nordic Food and Diet? Who developed this plan? How does it align with current dietary recommendations (or perhaps this diet is already recommended)?
Line 69: Because you’re specifically comparing a Nordic diet with the regular Swedish complementary diet the introduction would be a good place to explain the differences.
Line 120: What is the rationale for starting with sweeter flavours before moving on to bitter and sour? (Also, these should be described as tastes, not flavours).
Line 290: You mention on page 4 that the NG diet promotes plant foods—Is the difference in fruit and vegetable intake between diets in part due to the simple fact that the NG diet contains more fruits and vegetables than the CG?
Additionally, do differences in fruit and vegetable intake between groups really have anything at all to do with the Nordic diet? Why should we not expect the same result among infants in the conventional group if they were exposed to the same portions? I presume difference is a result of repeated exposure.
Line 369: What is a portfolio design?
Author Response
Dear Reviewer,
Thank you for the comments and suggestions to improve our manuscript. In the attached PDF file you find a point-by-point respond to your comments.
